# S100A8 and S100A9 in Hematologic Malignancies: From Development to Therapy

**DOI:** 10.3390/ijms241713382

**Published:** 2023-08-29

**Authors:** Farnaz Razmkhah, Sena Kim, Sora Lim, Abdul-Jalil Dania, Jaebok Choi

**Affiliations:** Division of Oncology, Department of Medicine, Washington University School of Medicine, St. Louis, MO 63110, USA; farnaz@wustl.edu (F.R.); senakim@wustl.edu (S.K.); slim22@wustl.edu (S.L.); a.d.dania@wustl.edu (A.-J.D.)

**Keywords:** S100A8, S100A9, calprotectin, signal transduction, hematologic malignancies

## Abstract

S100A8 and S100A9 are multifunctional proteins that can initiate various signaling pathways and modulate cell function both inside and outside immune cells, depending on their receptors, mediators, and molecular environment. They have been reported as dysregulated genes and proteins in a wide range of cancers, including hematologic malignancies, from diagnosis to response to therapy. The role of S100A8 and S100A9 in hematologic malignancies is highlighted due to their ability to work together or as antagonists to modify cell phenotype, including viability, differentiation, chemosensitivity, trafficking, and transcription strategies, which can lead to an oncogenic phase or reduced symptoms. In this review article, we discuss the critical roles of S100A8, S100A9, and calprotectin (heterodimer or heterotetramer forms of S100A8 and S100A9) in forming and promoting the malignant bone marrow microenvironment. We also focus on their potential roles as biomarkers and therapeutic targets in various stages of hematologic malignancies from diagnosis to treatment.

## 1. Introduction 

S100A8 and S100A9 (also known as myeloid-related proteins 8 and 14, respectively) are Ca^2+^-binding proteins belonging to the evolutionarily conserved S100 leukocyte proteins family in both mice and humans [1,2]. S100A8, S100A9, and their heterodimer or heterotetramer forms, which are called calprotectin (S100A8/A9), can be secreted from immune cells, such as neutrophils, monocytes, activated macrophages, and dendritic cells, especially during inflammation whereas in healthy conditions, they are secreted significantly less [3,4,5,6]. Heterodimeric S100A8/A9 can bind to Ca^2+^ (and Zn^2+^ and possibly other metal ions) via their α-helix motifs to form heterotetramers. Achieving this conformation, which is more stable than heterodimer and homodimer forms, enables them to interact with other proteins, transfer Ca^2+^ signals to them, and induce their structural conformation, thereby affecting their activity [7,8]. 

S100A8 and S100A9 are cytosolic proteins (which make up around 45% of total cytosolic proteins in neutrophils) that can also be released into the extracellular environment independently from granules and act as cytokines in both autocrine and paracrine manners [4,9]. Following their release, they are recognized by pattern recognition receptors (PRRs), such as Toll-like receptor-4 (TLR-4) and receptors for advanced glycation end products (RAGE) on the target cells [10,11,12]. The interaction with TLR-4 initiates a signaling cascade that mediates inflammation, tumor development, cell proliferation, and differentiation. In addition, they can interact with RAGE, resulting in the mobilization and migration of neutrophils, monocytes, and macrophages by inducing pro-inflammatory cytokines. Both of these two receptors’ pathways lead to the activation of nuclear factor kappa B (NF-kB) signaling pathway mediators [12,13,14]. In addition to their cytokine activity, S100A8 and S100A9 have a recently discovered role as transcription coactivators when localized in the nucleus. They can be recruited to promoters and enhancers during the transformation stage of breast cancer, thereby increasing the efficiency of oncogenic transformation [15]. 

The function of immune cells is impacted by S100A8 and S100A9 proteins. Neutrophils migrate to the site of inflammation or infection faster than other immune cells. At the time of neutrophil extracellular traps (NETs) formation (NETosis), which kills the pathogen extracellularly, S100A8/A9 is released from neutrophils and can activate them in the presence of 10% human serum by upregulating CD11b and adhesion marker (CD62L) [16]. S100A8 and S100A9 can also affect leukocyte trafficking and recruitment through the p38 mitogen-activated protein kinase (MAPK) signaling pathway. The reorganization and polymerization of microtubules that control leukocyte migration depend on phosphorylated S100A9 by MAPK in a calcium-dependent manner. Phosphorylated S100A9, when bound to S100A8, decreases microtubule formation. Meanwhile, S100A8 directly binds to tubulin and promotes its polymerization. It seems that S100A8 is the effector subunit of this complex, whereas S100A9 plays a regulatory role [17]. 

Hematologic malignancies (HMs) include a group of diseases with numerous and various genetic and epigenetic modifications which make them unique from each other in diagnosis to treatment. Based on the lineage framework, they are divided into myeloid and lymphoid neoplasms. Some of them are caused by uncontrolled proliferation (myeloproliferative neoplasms; MPN) and others by defects in normal differential and functional hematopoiesis (myelodysplastic syndromes; MDS), while leukemia is affected by both excess proliferation and arrested differentiation. Despite differences in their biogenesis mechanisms and whole genetics, they all share a common feature: the bone marrow (BM) microenvironment is the home of their birth and growth, where they bear genetic and epigenetic changes and interact with each other as well as with healthy hematopoietic and non-hematopoietic cells [18,19,20]. 

Most cancers (around 90%) are caused by somatic mutations and environmental factors such as chronic infections, smoking, and dietary factors. Many of these environmental factors are associated with some chronic inflammation [21]. Therefore, it is supposed that inflammation persists in the context of a multitude of cancers including HMs. Here, we review the multi-functional role of S100A8 and S100A9 in the route of HMs development, diagnosis, and treatment. 

## 2. S100A8 and S100A9 as a Biomarker in HMs

Several studies highlighted the upregulation of S100A8 and S100A9 in many cancers such as colorectal, gastric, prostate, melanoma, head and neck, and breast cancers [22,23,24,25,26,27,28]. Salama et al. reviewed reports published over a period of 46 years and concluded that the S100 family of proteins’ upregulation is commonly observed in a variety of tumors and is linked to tumor progression [29]. However, S100A9 functions as both a tumor suppressor and a tumor promoter [29]. During healthy myelopoiesis, vast S100A8 and S100A9 gene expression levels are observed from low levels in myeloid progenitor cells to high levels in differentiating granulocytes and monocytes [30,31]. However, S100A8 and S100A9 can have variable expression patterns during the malignant stage. For example, chronic lymphocytic leukemia (CLL) starts from an indolent phase and then progresses to an aggressive type. The disease remains stable for years without any treatment during the indolent phase. While progressing aggressively, oxidative stress, phosphatidylinositol 3-kinase (PI3K)/AKT, the NF-kB pathway, and inflammation are all activated in CLL cells [32]. S100A9 is overexpressed inside the CLL patients’ B cells with an aggressive form (Figure 1), as an activator of the NF-kB pathway [32]. This protein is also increased in the plasma exosomes of those patients [32]. Those exosomes containing S100A9 can activate the NF-kB pathway inside CLL cells [32]. Thus, increased levels of S100A9 inside both CLL cells and exosomes activate NF-kB signaling pathway mediators, which results in leukemic cell proliferation and survival signaling [32]. EMMPRIN, as a receptor of S100A9 [33], is also expressed on CLL leukemia cells as well as the T cells and macrophages/monocytes of healthy donors but not on normal B cells [32]. In CLL, the progression phase is correlated with S100A9 overexpression, which distinguishes it from the indolent phase. Thus, the time of this gene expression and cell distribution may even vary in a single disease. 

Variable levels of S100A8 and S100A9 in serum, plasma, or feces can be potential biomarkers of disease initiation, progression, response to therapy, and phenotype change. S100A8/A9 has been identified to be a biomarker in the inflammatory stages of numerous diseases such as ulcerative colitis and acute respiratory infections [34,35,36,37,38,39]. In addition, plasma levels of S100A8/A9 can be a surrogate of blood neutrophil count (below 1 × 10^9^/L) as a cost-effective, easier, and at-home method for monitoring and evaluating patients compared with flow cytometry [40]. Moreover, a study using quantitative protein expression profiling concluded that upregulated S100A8 in serum can be a diagnostic biomarker in two groups of childhood leukemia (B and T cell acute lymphoblastic leukemia; B and T-ALL) [41]. Another study showed that patients with Hodgkin lymphoma have a lower level of serum S100A8/A9 after treatment compared with pretreatment [42]. The latter study also examined inflammation markers within groups and found no correlation. Lower levels of S100A8/A9 post-treatment, which are independent of inflammation markers, may be a biomarker to predict a favorable response to treatment independent of inflammation condition [42]. On the other hand, elevated serum levels of S100A9 in patients with advanced extranodal NK/T cell lymphoma (ENKL) are correlated with an unfavorable response to pegaspargase/gemcitabine (chemoresistance) and poor overall survival [43]. 

Among the BM failure syndromes, aplastic anemia (AA) and MDS are characterized by progressive peripheral blood cytopenia. Investigating circulating S100A8, S100A9, and S100A8/A9 in patients’ plasma revealed an elevated level of S100A8 in MDS, but not in AA patients at the time of diagnosis, making it a differential diagnosis marker for these two diseases [44]. Furthermore, increased levels of S100A8/A9 in the BM plasma of MDS patients are associated with poor prognosis [45]. S100A8 is also upregulated (using proteomic profiling) in the BM mononuclear cells (MNCs) of acute myeloid leukemia (AML) patients at diagnosis, which is correlated with the worst prognosis [46]. Even though S100A8 is a granulocyte maturation marker and not expressed in normal bone marrow CD34+ HSCs, Nicolas et al. showed, in a retrospective study, that BM MNCs of AML patients have elevated levels of S100A8 as a predictor of low survival [46]. Besides the common predictors of prognosis in AML patients, such as cytogenetic abnormalities, S100A8 has been introduced as a new biomarker of prognosis [46]. 

Philadelphia-negative MPNs, including polycythemia vera (PV), primary myelofibrosis (PMF), and essential thrombocythemia (ET), are a heterogeneous group of hematopoietic stem cell diseases characterized by monoclonal feature and activated Janus kinase/signal transducers and activators of transcription (JAK/STAT) signaling, which results in chronic inflammation, the expansion of neoplastic clones, and the development of BM fibrosis [47]. Krecak et al. showed an elevated level of S100A8/A9 in all PV, ET, and PMF patients’ serum when compared to the control group and no significant differences in S100A8/A9 levels among these three groups [48]. They found that the sensitivity and specificity of S100A8/A9 are higher among PV, ET, and PMF patients than the well-known inflammatory marker, C reactive protein (CRP), which makes S100A8/A9 a useful diagnostic biomarker. However, other inflammatory disorders such as rheumatoid arthritis and systemic lupus erythematosus should be ruled out during the diagnosis of PV, ET, and PMF, since circulating S100A8/A9 is highly correlated with the inflammatory state of those inflammatory disorders as well [48]. Table 1 summarizes S100A8 and S100A9’s function as biomarkers in HMs. 

Overall, increased levels of S100A8, S100A9, and S100A8/A9 are correlated with the diagnosis and prognosis of HMs. Also, they can be negatively correlated with response to therapy. Interestingly, S100A8/A9 is upregulated in the inflammation phase of HMs more sensitively than common inflammation markers such as CRP.

## 3. Role of S100A8 and S100A9 in the BM Microenvironment of HMs

BM is the primary site of hematopoiesis (niche), where hematopoietic stem cells (HSCs) reside, proliferate, differentiate, and make the hematopoietic cell pool [49]. In this microenvironment, inflammation contributes to tumorigenesis by initiating and progressing the tumor directly and forming the tumor microenvironment (TME) indirectly [50]. Mechanisms include blocking anti-tumor immunity, shaping a tumor-permissive state through TME, and applying tumor-promoting signals on both epithelial and cancer cells [50]. These are mediated by immune cells (such as myeloid-derived suppressor cells; MDSCs) attracted to the tumor cells which support their niche with growth factors such as VEGF and EGF or by suppressing other immune cells such as Th1 cells [50,51]. An interesting study conducted by a Swedish group, comprising 9219 patients with AML, 1662 patients with MDS, and 42,878 matched controls, showed that autoimmune diseases which keep the immune system activated for a long time (chronic stimulation) can increase the risk of AML and MDS [52]. 

S100A8 and S100A9 upregulation can change the disease phenotype within the microenvironment (Figure 2). Mesenchymal stromal cells (MSCs), one of the most important niche-forming cells in the BM microenvironment, can communicate with AML cells through soluble factors (IL-6), which results in the upregulation of S100A8 and S100A9 expression (S100A8^high^ and S100A9^high^) and activation of the JAK/STAT3 signaling pathway in AML cells (Figure 2A). This S100A8^high^ and S100A9^high^ population shows increased cell differentiation (elevated maturation markers such as CD14 and CD11b) and chemoresistance compared to the S100A8^low^ and S100A9^low^ population [53]. Moreover, inflammation in the MSCs results in pre-leukemia evolution studied in Shwachman–Diamond syndrome (Figure 2B). The activation of the p53 pathway in CD271+ MSCs is accompanied by the overexpression of eleven genes including *S100A8* and *S100A9.* Then, S100A8 and S100A9 are secreted as S100A8/A9, which drives p53-S100A8/A9-TLR signaling in hematopoietic stem and progenitor cells (HSPC) as genotoxic stress-inducing mitochondrial dysfunction, oxidative stress, and activation of DNA damage response [54]. Leimkühler et al. showed that two MSC subsets (MSC1 and 2) are present in the BM microenvironment of MPNs. MSC1/2 can be reprogrammed to initiate differentiation in the extracellular matrix (ECM) secreting cells, such as myofibroblasts in the fibrotic stage, while losing their supporting role for hematopoiesis. Then, myofibroblasts form myelofibrosis by upregulating and depositing ECM proteins such as collagen, glycoproteins, and core matrisome in the BM (Figure 2C). In this study, S100A8/A9 upregulation in MSC1 and 2 proceeds the BM fibrosis as MPN phenotype [55]. Another study by Gleitz et al. showed that the overexpression of CXCL4 (also known as chemokine platelet factor-4) in hematopoietic cells runs the fibrosis in a PMF murine model, while CXCL4 deficiency reverses BM fibrosis. They reported that CXCL4 deficiency ameliorates the JAK/STAT signaling pathway in both megakaryocyte and stromal cells due to decreased levels of interferon-inducible target genes such as *Ifi27*, *Ifi35*, *Usp18*, and *Ifitm1* as well as *Stat1* [56]. These two studies point out that BM fibrosis is controlled by stromal cells in the BM microenvironment through both the levels of S100A8/A9 proteins and the activation of the JAK/STAT signaling pathway (as the arms of inflammation). 

MDSCs are another set of important BM microenvironment-constituting cells that can inhibit immune cells through multiple mechanisms. They expand themselves by upregulating STAT3, inhibit Th1 cells proliferation through arginase 1, peroxynitrite, and ROS induction, and induce regulatory T cells (Tregs). Interestingly, the upregulated STAT3, during MDSC expansion, can induce the expression and secretion of S100A8/A9 from myeloid progenitor cells. Since these proteins have receptors on the MDSCs, they can decrease differentiation while increasing the expansion of MDSCs [57,58]. These cells can also expand inside the BM microenvironment of MDS patients by an increased level of S100A9 (due to inflammation) (Figure 2D). S100A9, in close collaboration with its receptor CD33 on the MDSCs, activates CD33′s immune-receptor tyrosine-based inhibition motif (ITIM) signaling which results in both the expansion of MDSCs and secretion of IL-10 and TGF-β. Then, an increased number of MDSCs plus suppressive cytokines can affect the quality of hematopoiesis by inducing death in HSPCs [59]. This effect is also observed in aged S100A9 transgenic mice (14–16 months) which show MDSCs accumulation within the BM niche [60]. These cells cause ineffective hematopoiesis, including cytopenia and multi-lineage cytological dysplasia, resulting in hematopoiesis modification and resembling MDS [59]. The molecular pathways underlying this effect include the overexpression of S100A9 and activation of c-Myc, which trigger the aberrant expression and activation of PD-1 on HSPCs and PD-L1 on MDSCs. PD-1 and PD-L1 overexpression subsequently result in hematopoietic cell death, hematopoiesis suppression, and BM failure [60]. 

Upon considering the aforementioned observations, BM-microenvironment-forming cells, such as MDSCs and MSCs, can mediate several intracellular signaling such as p53, JAK/STAT, and CD33, resulting in S100A8 and S100A9 overexpression or vice versa. In this microenvironment, S100A8, S100A9, and S100A8/A9 promote HMs progression by modifying the disease phenotype to an aggressive form, inducing DNA damage, chemoresistance, and ineffective hematopoiesis, while suppressing the immune system through MDSCs. 

## 4. Role of S100A8 and S100A9 in the Treatment of HMs

As discussed in previous sections, S100A8/A9 can activate the various intracellular signaling pathways which affect cell fate. In some HMs, their expression levels are correlated with the progression phase (such as CLL), while in others, they are dysregulated at the time of diagnosis and predict prognosis (such as AML). Since S100A8 and S100A9 play a role in the pathogenesis and progression of HMs, they can be considered as potential therapeutic targets. In this way, S100A8 and S100A9 can be directly marked. Also, their upstream and downstream mediators and regulators can be targeted to indirectly modulate their expression and function. For example, a recent study has shown that the C/EBPδ transcription factor is a key regulator of *S100a8* and *S100a9* expression, which specifically binds to their promoter regions and changes their expression. Therefore, targeting C/EBPδ may modulate the inflammation stage of diseases mediated by *S100a8* and *S100a9* expression [61].

Although, S100A8 and S100A9 may act against each other. In the study by Laouedj et al., S100A9 alone can increase the differentiation of AML cells, which can be prevented by S100A8. Using recombinant S100A9 protein to make it dominant over S100A8 (high ratios of S100A9 to S100A8) in an AML mouse model resulted in increased AML cell maturation and mice survival (Figure 3). This effect was again observed when they used anti-S100A8 antibody to decrease S100A8. Thus, they concluded that a high ratio of S100A9 to S100A8 can promote AML cell differentiation and maturation. In this study, recombinant S100A9 protein binds to the TLR4 on the AML cells (in vitro) and leads to the phosphorylation and activation of p38 mitogen-activated protein kinase, extracellular signal-regulated kinases1/2 (ERK1/2), and c-Jun N-terminal kinase (JNK) signaling pathways, which result in AML cell differentiation [62].

S100A8 and S100A9 can affect the viability of leukemia cells, and thus be potential therapeutic targets (Figure 4). Lee et al. demonstrated the apoptotic role of S100A8/A9 in chronic eosinophilic leukemia (CEL) cell line (EoL-1) containing the *Fip1like 1-platelet derived growth factor receptor alpha* (*FIP1L1-PDGFRα)* fusion gene. The study showed that recombinant S100A8 and S100A9 proteins (5 and 10 ug/mL, after 48 and 72 h) induce apoptosis by modulating the Bcl family of proteins and the surface expression of TLR-4, as well as the caspase 3/9 pathway, in a time- and dose-dependent manner. In addition, they observed a reduction in FIP1L1-PDGFRα-mediated signaling by decreasing *FIP1L1-PDGFRα* gene expression in the presence of recombinant S100A8 and S100A9 [63] (Figure 4A). Another group showed that the in vitro administration of human recombinant S100A8/A9 (10, 20, 40, 80, and 100 mg/mL) on Nalm6 leukemia cell line induces an apoptotic effect in a time- and dose-dependent manner (after 24, 48, and 72 h) [64] (Figure 4B). A study by Kim et al. also revealed the pro-apoptotic effect of S100A8 and/or S100A9 (10 ug/mL for 48 and 72 h) on human monocytic leukemia cells (THP-1) which express high levels of TLR-4 [65] (Figure 4C). Moreover, the treatment of all-trans retinoic acid receptor α (ATRA), the most common therapy for APL, induces apoptosis and inhibits cell growth in acute promyelocytic leukemia (APL) cell lines (NB4 and PR9). This effect is linked to *PML/RARα* translocation as the most well-known cytogenetic signature of APL, enhanced with PU.1 transcription factor. Subsequently, PU.1 binds to the S100A9 promoter, upregulating *S100A9* expression, which leads to apoptosis in NB4 cells [66] (Figure 4D). 

In contrast, some studies demonstrate the anti-apoptotic role of S100A8 and S100A9, which induce drug resistance in leukemia cells. Yang et al. showed that the AML cell line (HL-60) transfected with S100A8 protein is resistant to etoposide (chemoresistance) and has less apoptosis due to altered transcription levels of apoptosis pathway mediators, such as Bcl-2, Bax, and Caspase-3 proteins [67] (Figure 5A). Another study showed that JQ1, a member of the Bromodomain and Extra-Terminal motif (BET) inhibitors, can induce apoptosis in AML cell lines (OCI-AML3 and THP-1) by suppressing S100A8 and S100A9 when synergistically applied with Daunorubicin [68] (Figure 5B). Curcumin, a biologically active polyphenol, can downregulate S100A8 in K562/DOX cells, a doxorubicin (DOX)-resistant chronic myelogenous leukemia (CML) cell line, which sensitizes these cells to DOX (Figure 5C). It has been shown that inhibiting S100A8 expression using small interfering RNA (si-S100A8) increases the K562/DOX cell line’s sensitivity to DOX through increased intracellular Ca^2+^ (in the absence of S100A8) and endoplasmic reticulum stress [69]. 

It remains unclear why S100A8 and S100A9 demonstrate the dichotomous outcomes (pro-apoptotic vs. anti-apoptotic), even though supplementing exogenous recombinant S100A8 and S100A9 proteins seems to induce pro-apoptotic effects. One possible explanation for these contrasting results is that the pro-apoptotic effect of S100A8 and S100A9 is observed when the target cells express TLR4, whereas their anti-apoptotic effect is observed only when HMs are treated with chemo agents. Thus, further investigations to elucidate the molecular mechanisms underlying the disparate effects of S100A8 and S100A9 are warranted. 

Multiple myeloma (MM) is characterized by the expansion of clonal and malignant plasma cells (myeloma cells) in the BM. A recent study by Liu et al. demonstrated a negative correlation between S100A8 and S100A9 and drug sensitivity in patients with MM treated with proteasome and histone deacetylase inhibitors (bortezomib and panobinostat) [70] (Figure 5D). The in vivo blockade of S100A9 interactions with TLR-4 using ABR-238901 on MDSCs in a mouse model of MM decreased the expression of inflammatory and pro-myeloma cytokines, such as IL-6 and IL-10, in MDSCs [71]. This inhibitor also reduced angiogenesis and, when combined with bortezomib, reduced tumor load compared with bortezomib alone, suggesting that extracellular S100A9 can induce MM progression [71] (Figure 6A). In addition, Meng et al. reported that myeloid-derived S100A9 (neutrophils and macrophages) could activate the NF-kB signaling pathway in myeloma cells by increasing tumor necrosis factor superfamily member 13b (TNFSF13B) signaling and thus promote the survival and proliferation of myeloma cells in both the mouse model of MM and patient primary myeloma cells [72]. The in vivo administration of paquinimod (ABR-215454), an S100A9 inhibitor, resulted in myeloma cell apoptosis [72] (Figure 6B). Therefore, targeting extracellular S100A9 may improve the treatment outcomes in MM patients.

As previously mentioned, MDSC promotes tumor progression by preventing the activation of Th1 cells. S100A8/A9 can bind to RAGE on MDSC, activate the NF-kB signaling pathway, and promote its migration. MDSC can also secrete S100A8/A9 that, again, binds to RAGE in an autocrine manner, which results in MDSC accumulation in the blood and secondary lymphoid organs of mice [73]. Thus, blocking this pathway may decrease the number of MDSCs. In addition, blocking S100A8/A9 using Tasquinimod can prevent fibrosis and MPN phenotype in a *JAK2V617F*-mutated murine model [55]. It has been shown that tasquinimod binds to S100A8/A9 and inhibits its interaction with TLR-4 and RAGE receptors [55]. The therapeutic effects of S100A8 and S100A9 in HMs are summarized in Table 2.

In summary, S100A8 and S100A9, in the presence of chemo agents, decrease leukemia cell apoptosis and induce chemoresistance. Therefore, targeting them can increase chemosensitivity and improve patients’ survival. On the other hand, their ability (in the absence of chemo agents) to induce AML cell differentiation and apoptosis in various groups of leukemia opens a new window for researchers in terms of how to manage these proteins to achieve the most benefit for patients.

## 5. Conclusions and Future Directions

In this article, we reviewed S100A8 and S100A9 proteins in the context of HMs, mostly focusing on BM microenvironment, diagnosis, and treatment. S100A8 and S100A9 may work together or separately, inside or outside cells. Their expression levels change during HMs’ progression and treatment, which makes them potential biomarkers for diagnosis, prognosis, and response to therapy. 

BM microenvironment cells, such as MDSCs and MSCs, promote HMs progression by upregulating S100A8 and S100A9. MDSCs that suppress the immune system are also expanded by S100A8 and S100A9 in the BM microenvironment. Thus, upregulated S100A8 and S100A9 in this microenvironment make a beneficial TME for HMs initiation and progression. In parallel, S100A8 and S100A9 reduce cell apoptosis and induce chemoresistance. Targeting these proteins or their receptors may be a promising therapeutic strategy for HMs management. In contrast, in the absence of chemo agents, S100A8 and S100A9 can improve HMs by inducing leukemia cell differentiation and apoptosis. Therefore, considering all aspects and functions of S100A8 and S100A9 is essential before using them as therapeutic targets. 

Future studies are warranted to investigate the differential expression of S100A8 and S100A9 in various stages of all HMs, especially in their niche. Then, they can be considered specific biomarkers or therapeutic targets for each HM. 

## Figures and Tables

**Figure 1 ijms-24-13382-f001:**
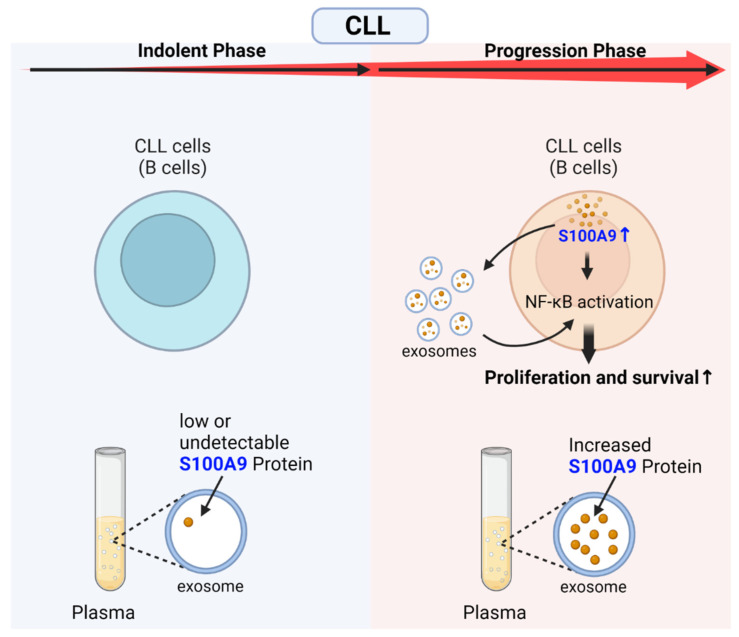
Progression model of the CLL through S100A9 overexpression in the leukemic cells and plasma exosomes. Increased expression of S100A9 within CLL leukemic cells during the disease progression (but not in the indolent phase) activates NF-kB signaling pathway mediators. In addition, secreted S100A9-containing exosomes from CLL leukemic cells activate the NF-kB signaling pathway mediators, which results in proliferation and survival signaling. Increased levels of S100A9 within the exosomes are detectable in CLL patients’ plasma during the disease progression. (CLL; chronic lymphocytic leukemia, NF-kB; nuclear factor kappa B). Created with BioRender.com (accessed on 18 August 2023).

**Figure 2 ijms-24-13382-f002:**
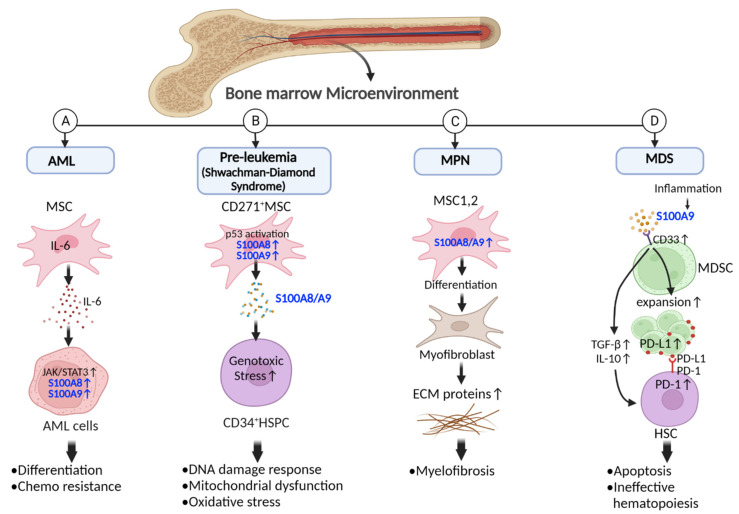
The function of BM microenvironment cells in the progression of HMs through S100A8 and S100A9. (**A**) Interaction between MSCs and AML cells through IL-6 results in increased levels of S100A8 and S100A9 which induce chemoresistance and differentiation of AML cells. (**B**) In the pre-leukemia model of Shwachman–Diamond syndrome, CD271+ MSCs overexpress S100A8 and S100A9 following activation of p53. Increased intracellular S100A8 and S100A9 results in increased secretion of S100A8/A9, which induces mitochondrial dysfunction, oxidative stress, and activation of DNA damage response in CD34+ HSPCs in the BM microenvironment. (**C**) In the pre-fibrotic stage of MPN, upregulated S100A8/A9 promote MSC1 and 2 differentiation into myofibroblasts, which secrete more ECM proteins and induce myelofibrosis. (**D**) In the BM microenvironment of MDS patients, inflammation leads to increased levels of S100A9, which bind to CD33 as a receptor on MDSCs, resulting in the expansion of MDSCs and increased secretion of TGFβ and IL-10. Aberrant expression of PD-1 on HSCs and PD-L1 on MDSCs in MDS patients besides the increased number of MDSCs and suppressive cytokines induce apoptosis and ineffective hematopoiesis in HSCs. (AML: acute myeloblastic leukemia, MSC: mesenchymal stromal cells, IL-6: interleukin 6, JAK/STAT3: Janus kinase/signal transducers and activators of transcription, HSPCs: hematopoietic stem and progenitor cells, MPN: myeloproliferative neoplasms, ECM: extracellular matrix, MDS: myelodysplastic syndromes, MDSC: myeloid-derived suppressor cell, HSC: hematopoietic stem cell, TGF-β: transforming growth factor beta, IL-10: interleukin 10, PD-1: programmed cell death protein 1, PD-L1: programmed cell death ligand 1). Created with BioRender.com.

**Figure 3 ijms-24-13382-f003:**
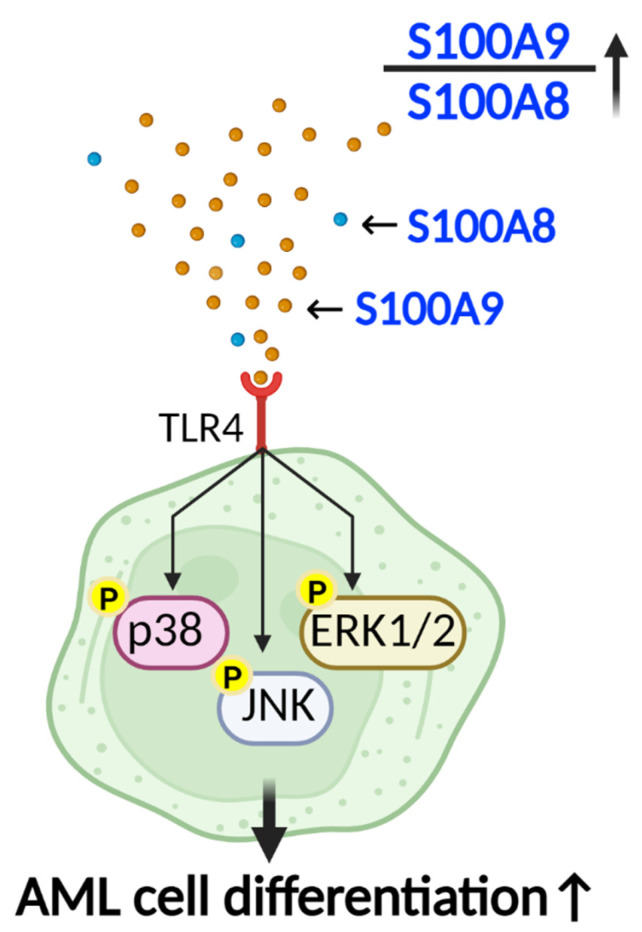
AML cell differentiation by S100A9. A high ratio of S100A9 to S100A8, when S100A9 binds to TLR-4, results in phosphorylation and activation of p38, JNK, and ERK1/2 signaling pathways. Activation of these pathways increases differentiation of AML cells and mice survival. (TLR4: toll-like receptor 4, ERK1/2: extracellular signal-regulated protein kinases 1 and 2, JNK: c-Jun N-terminal kinase, AML: acute myeloid leukemia). Created with BioRender.com.

**Figure 4 ijms-24-13382-f004:**
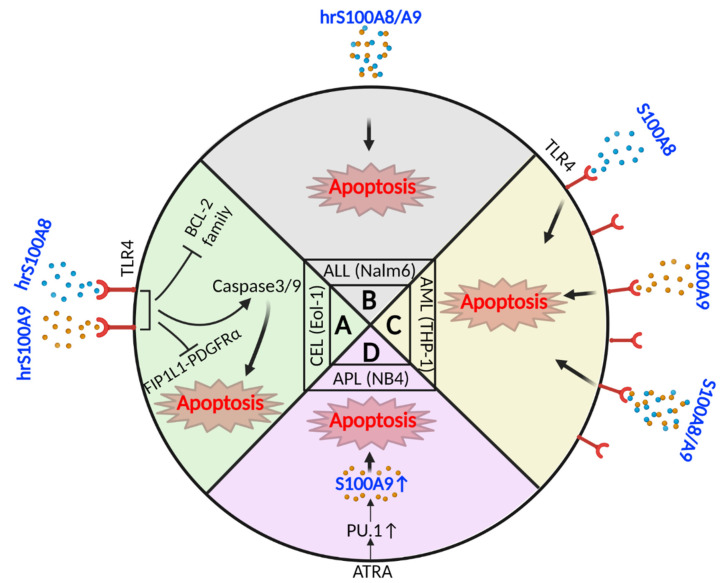
Leukemia cell apoptosis by S100A8 and S100A9. (**A**) Human recombinant S100A8 and S100A9 proteins separately bind to TLR4 on CEL cells and induce apoptosis by activating caspase 3/9 while downregulating BCL-2 family and FIP1L1-PDGFRα proteins. (**B**) Human recombinant S100A8/A9 can induce apoptosis in ALL cells. It remains unclear if S100A8/A9 functions intracellularly to induce apoptosis or extracellularly through binding to a receptor. (**C**) THP-1 cells, an AML M5 cell line, show upregulation of TLR4 on the leukemic cells. S100A8, S100A9, and S100A8/A9 all bind to this receptor and induce apoptosis. (**D**) In APL cells, ATRA restores PU.1 transcription factor, which increases S100A9 expression, resulting in cell apoptosis. It remains unclear if S100A9 functions intracellularly to induce apoptosis or extracellularly (secreted) through binding a receptor. (hrS100A8: human recombinant S100A8, hrS100A9: human recombinant S100A9, TLR4: toll-like receptor, FIP1L1-PDGFRα: Fip1like 1-platelet derived growth factor receptor alpha, BCL-2: B cell lymphoma 2, CEL: chronic eosinophilic leukemia, ALL: acute lymphoblastic leukemia, AML: acute myeloid leukemia, APL: acute promyelocytic leukemia, ATRA: all-trans retinoic acid receptor alpha). Created with BioRender.com.

**Figure 5 ijms-24-13382-f005:**
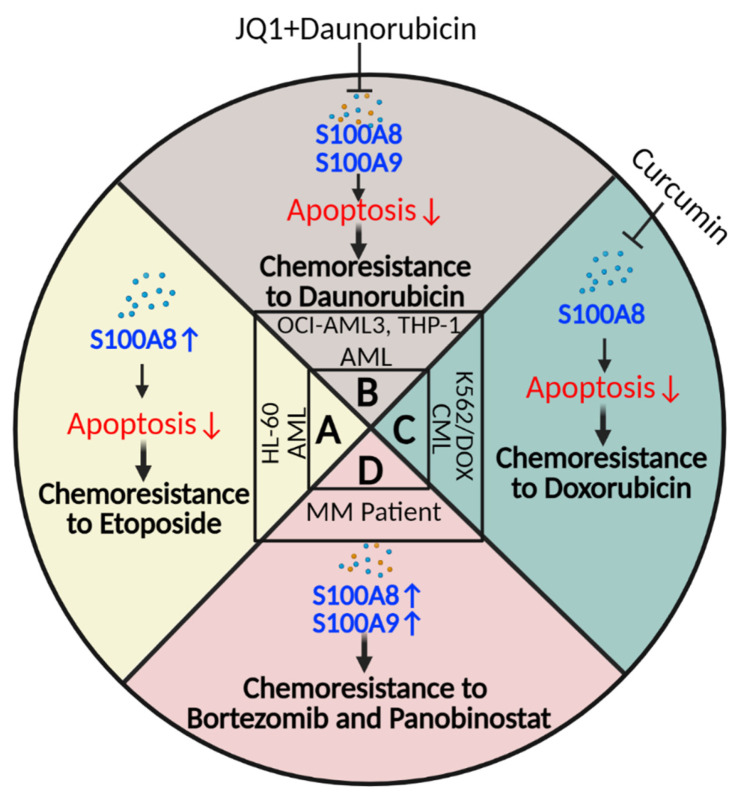
Chemoresistance by S100A8 and S100A9 in HMs. (**A**) Increased levels of S100A8 in HL-60 cells by transfection induce chemoresistance to etoposide by decreasing apoptosis. (**B**) JQ1 induces chemosensitivity to daunorubicin in AML patients and cell lines by suppressing S100A8 and S100A9 while increasing apoptosis. (**C**) Inhibiting S100A8 by si-RNA or curcumin increases chemosensitivity to doxorubicin in DOX-resistant CML cell line. (**D**) Increased levels of S100A8 and S100A9 in MM patients are correlated with chemoresistance to bortezomib and panobinostat. (AML: acute myeloid leukemia, CML: chronic myeloid leukemia, DOX: doxorubicin, MM: multiple myeloma). Created with BioRender.com.

**Figure 6 ijms-24-13382-f006:**
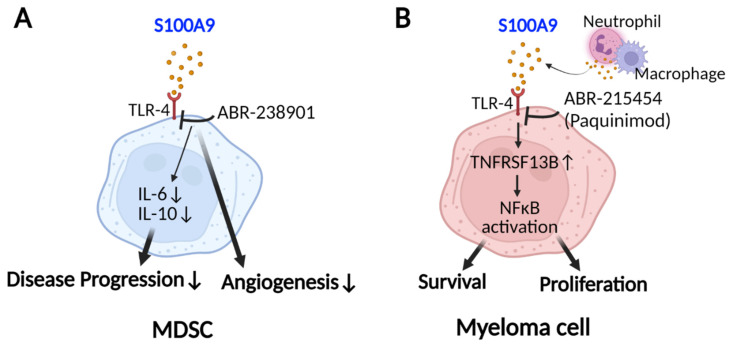
Effect of S100A9 blockade on MM management. (**A**) In vivo administration of ABR-238901 inhibits the interaction of extracellular S100A9 with its receptor (TLR4) on MDSCs, resulting in decreased secretion of IL-6 and IL-10, angiogenesis, and thus disease progression. (**B**) Neutrophil and macrophage-derived S100A9, when bound to TLR4 on myeloma cells, increases expression levels of TNFRSF13B and thus TNFSF13B signaling, which results in activation of NF-kB signaling, cell proliferation and survival. Paquinimod induces apoptosis in myeloma cells and decreases MM progression. (MDSC: myeloid-derived suppressor cell, TLR-4: toll-like receptor-4, IL-6: interleukin 6, IL-10: interleukin 10, TNFSF13B: tumor necrosis factor superfamily member 13b, NF-kB: nuclear factor kappa B). Created with BioRender.com.

**Table 1 ijms-24-13382-t001:** S100A8 and S100A9 as biomarkers in HMs.

Name	Source	Up/Down	Disease	Biomarker Role	Reference Number
S100A8	Serum	↑	Childhood ALL	Diagnosis	[41]
S100A8/A9	Serum	↓	Hodgkin Lymphoma	Response to therapy ↑	[42]
S100A9	Serum	↑	ENKL	Response to therapy and prognosis ↓	[43]
S100A8	Plasma	↑	MDS	Diagnosis	[44]
S100A8/A9	BM plasma	↑	MDS	Poor prognosis	[45]
S100A8	MNCs (BM)	↑	AML	Poor prognosis	[46]
S100A8/A9	Serum	↑	PV, ET, PMF	Diagnosis (Inflammatory state)	[48]

(ALL: acute lymphoblastic leukemia, ENKL: extranodal NK/T cell lymphoma, MDS: myelodysplastic syndromes, AML: acute myeloid leukemia, PV: polycythemia vera, ET: essential thrombocythemia, PM: primary myelofibrosis).

**Table 2 ijms-24-13382-t002:** Therapeutic effects of S100A8 and S100A9 in HMs.

Disease Name	Therapeutic Effect	Caused by	Reference Number
AML	Cell differentiation	S100A9 (when dominants over S100A8)	[62]
CEL	Apoptosis	S100A8 ↑ S100A9 ↑	[63]
ALL	Apoptosis	S100A8/A9 ↑	[64]
AML(M5)	Apoptosis	S100A8 ↑ S100A9 ↑ S100A9/A9 ↑	[65]
AML(M3)	Apoptosis	S100A9 ↑	[66]
MM	Decrease in inflammatory and pro-myeloma cytokines, inducing apoptosis	S100A9 ↓	[71,72]
MPN	Prevention of fibrosis and MPN phenotype	S100A8/A9 ↓	[55]

(AML: acute myeloid leukemia, CEL: chronic eosinophilic leukemia, ALL: acute lymphoblastic leukemia, MM: multiple myeloma, MPN: myeloproliferative neoplasm).

## Data Availability

No new data were created in this study. All the data summarized in this review were from original articles that were cited in the main text.

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
