# Peer review of "S100A8 and S100A9 in Hematologic Malignancies: From Development to Therapy"

_ijms, 2023, doi:10.3390/ijms241713382_

Round 1

Reviewer 1 Report

The Manuscript is a comprehensive review of the role of S100A8/S100A9 in hematologic malignancies. The review is well written and clearly shows the importance of these molecules in pathology and therapy. To be further improved, the authors could add to Figures and Tables the explanation of abbreviations, as not all readers may be used to all abbreviations used in Figures and Tables.

Author Response

To reviewers of Manuscript #ijms-2564799:

We appreciate and thank the reviewers for their helpful comments on our manuscript entitled “S100A8 and S100A9 in hematologic malignancies: from development to therapy” which definitely improve the quality of this manuscript. The revised version has been prepared based on the received comments from reviewers (highlighted in yellow). We hope the revised manuscript will be suitable for publication in IJMS. 

Reviewer #1:

The Manuscript is a comprehensive review of the role of S100A8/S100A9 in hematologic malignancies. The review is well written and clearly shows the importance of these molecules in pathology and therapy. To be further improved, the authors could add to Figures and Tables the explanation of abbreviations, as not all readers may be used to all abbreviations used in Figures and Tables.

Our response: We thank the reviewer for this comment, and we agree that an explanation of abbreviations inside figures and Tables can be very helpful for readers. We have added the explanation of all abbreviations used in the Figures and Tables (highlighted in yellow). We have also corrected one typo in line 317: “AML patients” to “AML cell lines (OCI-AML3 and THP-1)” – Fig. 5B has been updated accordingly.

Reviewer 2 Report

This paper is a review that discusses the expression, clinical significance, and potential applications of S100A8/S100A9 in hematopoietic tumors, citing recent papers. It is reasonably informative. However, the writing style is descriptive, which can be confusing for readers when, for example, a statement about the involvement of S100A8/S100A9 in the progression of hematopoietic malignancies (HMs) is followed by a description of how S100A8/S100A9 induces apoptosis in leukemia cells. It requires sufficient explanations, such as experimental conditions (concentration and time), to clarify why S100A8/S100A9, which contributes to the progression of leukemia, also exhibits leukemia-inhibitory effects. Another concern is the description about GVHD and microbiota, because these topics are heterogeneous compared to the main theme of this review. I think the authors should focus on HMs and the BM microenvironment. So regarding to Table 1, P8, and Figure 4, as well as the description in Section 5, removing them would make the review more coherent and understandable.

Author Response

To reviewers of Manuscript #ijms-2564799:

We appreciate and thank the reviewers for their helpful comments on our manuscript entitled “S100A8 and S100A9 in hematologic malignancies: from development to therapy” which definitely improve the quality of this manuscript. The revised version has been prepared based on the received comments from reviewers (highlighted in yellow). We hope the revised manuscript will be suitable for publication in IJMS. 

Reviewer #2:

This paper is a review that discusses the expression, clinical significance, and potential applications of S100A8/S100A9 in hematopoietic tumors, citing recent papers. It is reasonably informative. However, the writing style is descriptive, which can be confusing for readers when, for example, a statement about the involvement of S100A8/S100A9 in the progression of hematopoietic malignancies (HMs) is followed by a description of how S100A8/S100A9 induces apoptosis in leukemia cells.

1) It requires sufficient explanations, such as experimental conditions (concentration and time), to clarify why S100A8/S100A9, which contributes to the progression of leukemia, also exhibits leukemia-inhibitory effects.

Our response: We appreciate the reviewer for the valuable comments. We agree with the reviewer. Thus, in this revision, we have added the following (lines 325-331), “It remains unclear why S100A8 and S100A9 demonstrate the dichotomous outcomes (pro-apoptotic vs. anti-apoptotic) even though supplementing exogenous recombinant S100A8 and S100A9 proteins seems to induce pro-apoptotic effects. One possible explanation for these contrasting results is that the pro-apoptotic effect of S100A8 and S100A9 is observed when the target cells express TLR4, whereas their anti-apoptotic effect is observed only when HMs are treated with chemo agents. Thus, further investigations to elucidate the molecular mechanisms underlying the disparate effects of S100A8 and S100A9 are warranted.” We have also corrected one typo in line 317: “AML patients” to “AML cell lines (OCI-AML3 and THP-1)” – Fig. 5B has been updated accordingly.

2) Another concern is the description about GVHD and microbiota, because these topics are heterogeneous compared to the main theme of this review. I think the authors should focus on HMs and the BM microenvironment. So regarding to Table 1, P8, and Figure 4, as well as the description in Section 5, removing them would make the review more coherent and understandable.

Our response: We appreciate the reviewer for this perspective. We agree that omitting the description about GVHD and microbiota from this manuscript could make it more coherent. Therefore, we have removed all contents regarding GVHD and microbiota from Table 1 and Figure 4, and the whole Section 5 as well as the related parts from the main text.

Round 2

Reviewer 2 Report

The manuscript has been well revised and I think, is acceptable for publication.